# Eliminating the Effect of Reflectance Properties on Reconstruction in Stripe Structured Light System

**DOI:** 10.3390/s20226564

**Published:** 2020-11-17

**Authors:** Zhao Song, Zhan Song, Yuping Ye

**Affiliations:** 1Shenzhen Institutes of Advanced Technology, Chinese Academy of Sciences, Shenzhen 518055, China; zhao.song@siat.ac.cn (Z.S.); yp.ye@siat.ac.cn (Y.Y.); 2University of Chinese Academy of Sciences, Beijing 100049, China; 3Mechanical and Automation Engineering Department, The Chinese University of Hong Kong, Hong Kong 3947700, China

**Keywords:** stripe structured light, photometric stereo, blinn-phong model, stripe detection, reflectance property

## Abstract

The acquisition of the geometry of general scenes is related to the interplay of surface geometry, material properties and illumination characteristics. Surface texture and non-Lambertian reflectance properties degrade the reconstruction results by structured light technique. Existing structured light techniques focus on different coding strategy and light sources to improve reconstruction accuracy. The hybrid system consisting of a structured light technique and photometric stereo combines the depth value with normal information to refine the reconstruction results. In this paper, we propose a novel hybrid system consisting of stripe-based structured light and photometric stereo. The effect of surface texture and non-Lambertian reflection on stripe detection is first concluded. Contrary to existing fusion strategy, we propose an improved method for stripe detection to reduce the above factor’s effects on accuracy. The reconstruction problem for general scene comes down to using reflectance properties to improve the accuracy of stripe detection. Several objects, including checkerboard, metal-flat plane and free-form objects with complex reflectance properties, were reconstructed to validate our proposed method, which illustrates the effectiveness on improving the reconstruction accuracy of complex objects. The three-step phase-shifting algorithm was implemented and the reconstruction results were given and also compared with ours. In addition, our proposed framework provides a new feasible scheme for solving the ongoing problem of the reconstruction of complex objects with variant reflectance. The problem can be solved by subtracting the non-Lambertian components from the original grey values of stripe to improve the accuracy of stripe detection. In the future, based on stripe structured light technique, more general reflection models can be used to model different types of reflection properties of complex objects.

## 1. Introduction

The acquisition of the geometry of a general scenario is one of the main research topics in computer vision and graphics. In particular, there is an increasing demand for high-precision and high-quality depth information of complex objects. As concluded in [1], the reconstruction quality of an object is strongly dependent on the interplay of surface geometry, material properties and incident illumination. As one of the active 3D shape acquisition techniques, structured light techniques, especially stripe-based structured light methods [2,3], can acquire micron-sized reconstruction accuracy for diffuse surfaces, whereas only degraded quality can be acquired for a non-Lambertian surface due to complex reflectance behavior. In detail, all structured light methods assume that the modulation of pattern originates from the surface geometry and the effect of reflectance properties on the profile of the stripe is either ignored or reduced by using different types of coding strategy [4] or light sources [5]. Recently, the main research interests in the field of structured light technique have been focusing on the coding strategy [6,7,8] and robust decoding method to improve reconstruction accuracy for a general scenario. Several hybrid methods [9,10] combining depth information from structured light technique and normal information from photometric stereo or shape from shading were proposed as well. The above-mentioned hybrid methods mainly focus on the refinement of geometric information based on normal information. Reflectance properties were not taken into account explicitly in the process of decoding. Essentially, those hybrid methods are an effective technology of information screening and fusion. Thus, the results of fusion strongly depend on the accuracy of individual systems and the effects of reflectance properties on reconstruction quality in structured light systems cannot be eliminated completely.

Apart from stripe-based line shifting strategy, phase shifting method is another main 3D reconstruction technique in the field of time multiplexing coding strategy, which combines Gray code or multiple-frequency patterns with phase-shifting sinusoidal fringe patterns. For a complex scene, compared with phase-shifting method, the main advantages of stripe-based structured light technique used in this paper are robustness and higher projection speed.

As concluded above, in this paper, we focus on the acquisition of the geometry of a general scenario with variant reflectance properties based on the hybrid system consisting of stripe structured light and photometric stereo. Differently from existing fusion methods, we explicitly formulate reflectance properties in the process of decoding to eliminate the effect of variant reflectance properties on the reconstruction quality in a structured light system. We start with the subpixel detection of stripe, which is crucial for stripe-based structured light techniques, and an error analysis of stripe detection caused by reflectance properties is concluded. Secondly, the estimated reflectance by photometric stereo was used explicitly in the process of stripe detection, which reduces the effect of reflectance properties on stripe detection effectively.

To the best of our knowledge, our work is the first to address the reconstruction problem of complex objects by combining stripe-based structured light technique with photometric stereo. We decompose the problem of complex scene reconstruction into the small problem of using reflectance information to optimize stripe detection. Based on the error analysis of stripe detection and our proposed method, a new feasible scheme is provided for the reconstruction of complex scenes. The flowchart of the proposed 3D measurement system is shown in Figure 1.

The main contributions of the paper are:(a)It proposes the first hybrid system consisting of stripe-based structured light technique and photometric stereo.(b)It proposes a model for the subpixel detection of stripe in the process of decoding, which takes the reflectance properties into account explicitly.

The remainder of this paper is organized as follows. Section 2 introduces the related work about time multiplexing structured light techniques, photometric stereo and the hybrid methods. Section 3 presents the error analysis of stripe-based structured light techniques. Section 4 proposes our methods to eliminate the effect of variant reflectance on stripe detection. Experiment results and the conclusion of our work are presented in Section 5 and Section 6, respectively.

## 2. Related Work

**Stripe structured light.** Initially, Gray code or Binary code [11,12] were used as coding strategies of stripe-based structured light technique. To maximize the minimum stripe width for the same coding capacity, Gray code with line shifting was proposed as shown in Figure 2. Recently, considering the different effects of global illumination on different frequency patterns, MinSW8 [13] and an alternative binary structured light pattern [14] were designed by simple logical operations and tools from combinatorial mathematics to improve accuracy and robustness for complex scenes. The above work improves the robustness and accuracy of stripe-based structured light technique by utilizing different coding strategies. In this paper, to maximize the minimum stripe width of the coding patterns and eliminate the blurring effect on reconstruction accuracy, we selected Gray code and line shift as our coding strategy.

As for decoding, stripe detection with subpixel accuracy is crucial. With the normal and inverse pattern projected or not, linear interpolation method [15] was proposed to calculate the subpixel location with captured stripe images. As shown in Figure 3a, the intersection point of the line *AB* with *CD* or the line *AB* with *EF* is defined as the stripe edge position *P*. This method assumes the modulation of the profile of stripe results from geometry and ignores the effect of surface texture on stripe detection completely. For diffuse surface with texture or non-Lambertian surface, reconstruction error caused by reflectance properties is introduced. To reduce the effect of texture on stripe detection, the normalization method was used. With all-white and all-black patterns projected, the grey value of stripe corresponding to each pixel (*i*, *j*) in the camera image was normalized firstly as follows:(1)$$\bar{I}(i,j)=\frac{I(i,j)-I_{0}(i,j)}{I_{255}(i,j)-I_{0}(i,j)}$$
where *I*(*i*, *j*), $\bar{I}(i,j)$ are the original and normalized grey value of the profile of stripe in camera images and $I_{255}(i,j)$, $I_{0}(i,j)$ are grey values with projected all-white and all-black patterns corresponding to pixel coordinate (*i*, *j*) in the camera image.

This method that assumes the relationship between reflectance property and the intensity of incident illumination is linear. For diffuse surface with texture, the reconstruction error of white paper with characters printed declines to 0.12 mm, while for non-Lambertian surface, the error still exists. As shown in Figure 3b, an improved zero-crossing feature detector was proposed to improve the reconstruction accuracy for shiny surface. Apart from linear interpolation method, the Gaussian function [16] can be used to model the blurred edge mathematically to present the profile of stripe with better accuracy at the cost of computation time. In this paper, for diffuse surface with texture and non-Lambertian surface with complex reflectance properties, we proposed a novel stripe detection method. Neither linear assumption nor ignoring the effect of reflection properties on reconstruction is needed. Our proposed method acquires the reflectance properties by photometric stereo followed by eliminating the non-Lambertian components in the process of stripe detection.

On close-range photometric stereo, several calibration and reconstruction methods [17,18,19,20] for diffuse surface were proposed. A general way to model surface reflectance has been studied for non-Lambertian surface ranging from BRDF to all kinds of simplified models [21]. In this paper, similar to the method in [22], we select the Blinn-Phong model [23] to model the reflectance properties.

**Phase-shifting structured light.***N*-step phase-shifting profilometry [11,24,25,26,27,28] is another commonly used technique for 3D measurements. In recent work, for textured surface, the method in [29] corrected the recovered phases by template convolution in 3×3 or 5×5 pixel windows. As the intensities of camera images reach the maximum intensity limitation of the camera sensor for shiny surface, a high dynamic range (HDR) 3D measurement technique were proposed in [25,30]. By changing the exposure time of the camera or generating adaptive fringe patterns, multiple projections are needed to reconstruct the shiny object. An adaptive fringe projection technique was proposed in [31].

**Hybrid methods.** In [32], three types of approaches were reviewed and discussed respectively, including fusion approaches [9,33], subsequent approaches [34,35] and joint approaches [32,36]. Without direct interaction during the acquisition process of the absolute depth and normal, the fusion approaches combine the depth values and normal information directly to improve the reconstruction results based on individual system. Subsequent approaches refine the poor point cloud with normal information. Joint approaches use global optimization frameworks modeling the reconstruction problem with all image inputs as constraint and acquire the reflectance, normal and depth values simultaneously by solving the non-linear convex optimization problem. Closely related to this paper is the work of Jakub Krzesłowski and Robert Sitnik [22] where phase shift and Gray code was used as a coding strategy and the specular attributes were utilized to correct the phase value of each pixel. A key difference is that it was not phase shift, but line shift that was used in our methods. Different coding strategy leads to different decoding strategy. In this paper, we propose methods to utilize reflectance properties to improve stripe detection accuracy instead of phase values.

## 3. Error Analysis on Stripe Detection

As concluded in Section 2, several coding patterns based on stripe edges can be used as coding strategies and the difference of binary stripe-based coding strategy lies in the number of patterns needed to be projected, as well as in the maximum and minimum stripe width for the same coding capacity. The subpixel accuracy of stripe detection is crucial for all stripe-based structured light methods. In this paper, we selected Gray code and line shift in Figure 2 as our coding strategy and the intersection between the lines from normal and inverse stripe patterns was defined as the subpixel value of stripe detection. In detail, as shown in Figure 3a, the position of the stripe edge *P* is defined as the intersection between the line *AB* and *EF* from the normal and inverse stripe patterns, respectively. Mathematically, with the captured images of normal and inverse stripe patterns, the zero-crossing points *L_P_* is first calculated with pixel accuracy. Then, with a predefined width parameter *n* centered at *L_P_*, the intensity vector ***I_p_*** and ***I_n_*** with 2*n* + 1 pixels can be acquired from the normal and inverse pattern images. The parameters $a^{*}$ and $b^{*}$ of the fitted line are acquired by least-square estimation analytically:(2)$$b^{*}=\frac{\sum_{i=1}^{2n+1}i\cdot I_{i}-(2n+1)\cdot avg\cdot \bar{I}}{\sum_{i=1}^{2n+1}i^{2}-(2n+1)\cdot avg^{2}}$$
(3)$$a^{*}=\bar{I}-b^{*}\cdot avg$$
where $avg=(\sum_{i=1}^{2n+1}i)/(2n+1)$ and $\bar{I}=(\sum_{i=1}^{2n+1}I_{i})/(2n+1)$.

With $a_{0}^{*}$, $b_{0}^{*}$ from the normal image and $a_{1}^{*}$, $b_{1}^{*}$ from the inverse image, we have the following equation:(4)$$a_{0}^{*}\cdot x+b_{0}^{*}=a_{1}^{*}\cdot x+b_{1}^{*}$$

The subpixel value *p* of stripe edge is
(5)$$p=L_{P}+\frac{b_{1}^{*}-b_{0}^{*}}{a_{0}^{*}-a_{1}^{*}}-(n+1)$$

Via the above method of stripe detection, for diffuse surface without texture, with the modulation of profile resulting from surface geometry, micron-sized reconstruction accuracy can be acquired. However, for complex objects with non-uniform or variant reflectance properties, different types of detection errors are introduced. In the following parts of this section, the errors are divided into two categories, i.e., biased and missed detection.

### 3.1. Biased Dectection

In this case, although the zero-crossing point can be acquired, the subpixel value is changed due to the non-uniform and variant reflectance, which leads to a biased reconstruction error within 1 mm. As shown in Figure 4a, a metal plane is imaged with an all-white pattern projected. The profile of the stripe is changed by variant reflectance, which leads to biased detection. We observe that with mixed diffuse and specular reflectance, the profile of the same stripe varies along the *y*-axis. The maximum error is up to 0.6 pixels near the region with specular reflectance, which introduces the maximum reconstruction error with 0.8 mm along *x*-axis. In Figure 4b, a colored plane with texture was imaged with all-white pattern projected. The profile of the stripe is changed by surface texture, which also results in biased reconstruction. The reconstruction results along the red line were fitted into a line and the errors were shown in Figure 4b. We can observe obvious unevenness near the boundary of the surface texture and the maximum error of reconstruction results is 0.8 mm. In Figure 4c, a vase with glossy reflection and colored texture was also reconstructed. Due to glossy reflection and surface texture, apparent concave and convex parts are observed in reconstruction results via stripe-based structured light technique.

### 3.2. Missed Dectection

In this case, the zero-crossing point cannot be acquired due to an incomplete profile caused by surface material or occlusion, which leads to the hole in the reconstruction surface.

As shown in Figure 5a (red box), the specular area is over-exposed with an all-white pattern projected. Thus, the coding information is missed, which results in the hole in the reconstruction results. Meanwhile, in Figure 5a (blue box), due to geometrical occlusion, the shadow also results in missed detection. The other example is shown in Figure 5b. Because the black area absorbs most of the incident light, the profile of the stripe is missed. The reconstructed surface has an apparent hole. In addition, for transparent or translucent objects, the incident light passes through the surface and little light is reflected, which also leads to missed detection. In this paper, we are not involved in the reconstruction problem of transparent or translucent objects.

In terms of stripe detection, taking all interplay of geometry, surface reflectance and illumination into consideration, there are two types of error, i.e., biased and missed detection. No matter how complicated the interplay is, we can acquire high-quality point cloud by improving the stripe detection. In the following section, we propose an improved stripe detection method to improve the robustness and accuracy of stripe detection and further acquire improved reconstruction results of complex objects by stripe-based structured light method.

## 4. Improving Detection Accuracy Based on Reflectance

Per the error analysis in Section 3, taking reflectance properties into consideration, an improved stripe detection method is proposed in this section.

### 4.1. Reflectance Model

For diffuse surface with texture, apart from geometry, the intensity of each pixel in the image is related to albedo; for non-Lambertian, apart from incident illumination, the intensity of each pixel in the image is related to observation direction. Thus, taking reflectance properties into consideration, we select the Blinn-Phong model [23] to present the reflectance properties. Firstly, by the calibration method in [37], each area light source can be represented by the intensity *E* and direction ***l***. For the Blinn-Phong model, not only the incident direction ***l*** but also the observation direction ***h*** is taken into consideration. Regarding the nonlinear response between the incident intensity *E* and the reflected intensity *I*, the model divides the reflected intensity *I* into two components with $\varphi_{d}$ as the diffuse coefficient and $\varphi_{s}$, *e* as the specular reflection coefficients. As shown in Figure 6a and Equation (6), for each incident direction ***l*** and the surface point P, a different diffuse part of the surface response adds up to the specular components based on the half-way angle ***h***. The half-way angle ***h*** can be calculated by the initial depth information from stripe-based structured light system and the calibration results of the system.

For *I*, the area light source with incident direction ***l_i_***, the reflected intensity *I_i_* of the surface point P is calculated as follows:(6)$$I_{i}=E_{i}\left(\varphi_{d}\left(n\cdot l_{i}\right)+\varphi_{s}{\left(n\cdot h_{i}\right)}^{e}\right)$$
where ***n*** and *E_i_* is the normal vector and the incident intensity of the area light source, respectively.

The halfway vector ***h_i_*** is defined as:(7)$$h_{i}=\frac{v_{i}+l_{i}}{\left\|v_{i}+l_{i}\right\|}$$
where ***v_i_*** is the observation direction and is derived from the initial depth value via a stripe structured light system.

For *m* light sources, the non-linear problem is written as:(8)$$\varphi_{d}\left[\begin{array}{l} l_{1}\\ \vdots \\ l_{m} \end{array}\right]\cdot n+\rho_{s}{\left(\left[\begin{array}{l} h_{1}\\ \vdots \\ h_{m} \end{array}\right]\cdot n\right)}^{e}=\left[\begin{array}{l} \frac{I_{1}}{E_{1}}\\ \vdots \\ \frac{I_{m}}{E_{m}} \end{array}\right]$$

The above non-linear problem can be solved by optimization toolbox [38]. The initial value of ***n*** and ***h*** can be acquired based on depth values via stripe structured light system.

### 4.2. Improving Stripe Detection Based on Reflectance

For each pixel in camera image, the optimal parameters $\varphi_{d}^{*}$, $\varphi_{s}^{*}$, $e^{*}$ and $n^{*}$ in Equation (8) can be acquired by the optimization method. Thus, after acquiring the zero-crossing point with the intensity vectors ***I_p_*** and ***I_n_***, we adjust the profile of the stripe from the normal and inverse image based on the above-estimated parameters.

Assuming $I_{n}^{j}$ to be the *j* intensity of ***I_n_*** with the optimal parameters $\varphi_{d}^{*}{}^{j}$, $\varphi_{s}^{*}{}^{j}$, $e^{*}{}^{j}$ and $n^{*}{}^{j}$, the reflected intensity $I_{n}^{j}$ can be rewritten as:(9)$$I_{n}^{j}=E_{n}^{j}\left(\rho_{d}^{*}{}^{j}(n^{*j}\cdot l_{p})+\rho_{e}^{*}{}^{j}{\left(n^{*j}\cdot h_{p}^{j}\right)}^{e^{*j}}\right)$$
where ***l_p_*** is the incident direction of the projector and ***h_p_*** is the half-way vector based on the calibration parameters and initial depth from structured light system.

Thus, the intensity $E_{n}^{j}$ is calculated firstly as:(10)$$E_{n}^{j}=I_{n}^{j}/\left(\rho_{d}^{*}{}^{j}(n^{*j}\cdot l_{p})+\rho_{e}^{*}{}^{j}{\left(n^{*j}\cdot h_{p}^{j}\right)}^{e^{*j}}\right)$$

Then the adjusted intensity value ${\tilde{I}}_{n}^{j}$ with geometry modulation only is calculated as:(11)$${\tilde{I}}_{n}^{j}=E_{n}^{j}\cdot \frac{(I_{n}^{j}/E_{n}^{j})-\rho_{e}^{*}{}^{j}{\left(n^{*j}\cdot h_{p}^{j}\right)}^{e^{*j}}}{\rho_{d}^{*}{}^{j}}$$

Finally, the subpixel position of the stripe is calculated based on the adjusted intensity vector ${\tilde{I}}_{n}$ and ${\tilde{I}}_{p}$ via Equations (2)–(5).

The illustration of our improved detection method is shown in Figure 6b.

## 5. Experiment and Discussion

This section presents reconstruction results of several objects by our proposed method and comparisons with the existing methods. A flat checkerboard and metal-flat plane were used firstly to show measurement accuracy and the improvement on stripe detection quantitatively. To demonstrate the effectiveness on a free-form object, a vase with colored texture and a metal object were also reconstructed. The original and improved reconstruction results were presented and compared to validate our proposed method.

Our hybrid system consists of a monochrome camera (Point Grey-Blackfly S, with a resolution of 2448 × 2048), an industrial projector (TI-DLP4500, with a resolution of 912 × 1140) and six area light sources. The six area light sources are placed on a circular plane centered on the camera. The camera and projector are triggered synchronously by the trigger wire. The camera and area light source are triggered by a single-chip system. The working distance of the system is 35 to 45 cm and the working range of the equipment is 40 × 30 cm. The calibration methods in [37,39,40,41] were used to calibrate structured light system and area light source, respectively. The hardware system and calibration results are shown in Figure 7. In addition, a three-step phase-shifting algorithm in [29] was implemented and the reconstruction results and comparisons were also given. The phase-shifting pattern is set to have a period of 32 pixels and be shifted twice. For each pixel (*x*, *y*) in the camera image, the phase value ϕ(x,y) can be calculated as:(12)$$\phi (x,y)=\tan^{-1}\left(\frac{\sqrt{3}\left(F_{1}-F_{2}\right)}{2F_{2}-F_{1}-F_{3}}\right)$$
where *F*_1_, *F*_2_ and *F*_3_ are the grey values for pixel (*x*, *y*) in camera images with three phase-shifting patterns projected. The ambiguity was solved by acquiring the absolute depth values with 5-bit Gray code patterns projected.

### 5.1. Plane

To demonstrate the improvement of our proposed method on stripe detection quantitatively, a checkerboard and metal plane were reconstructed, respectively. The results from the same row in the camera image were fitted by a line and the error was used to evaluate the reconstruction accuracy. We evaluate of the performance of our method by numerical comparisons with the existing stripe-based structured light methods [2,42]. Here, for diffuse surface with texture, our method is compared with the normalization method by Gühring et al. [42]. For the metal plane, our method is compared with the methods in [2].

Firstly, a checkerboard with texture was reconstructed to verify that our proposed method would reasonably reduce the effect of texture on stripe detection. The results are illustrated in Figure 8 and Table 1. The normalization method cannot eliminate the effect of surface texture on stripe detection. An obvious error near the boundary is introduced with the maximum error 0.5077 mm and the minimum error −0.5781 mm. For a checkerboard plane, the overall error within 1 mm can be observed. To our best of knowledge, all existing stripe-based structured light decoding methods cannot eliminate it completely. By our improved detection method, the maximum and minimum error decreased to 0.15 and −0.15 mm with the overall error lower than 0.3 mm. As shown in Figure 8b, a smooth surface can be acquired. In addition, the reconstruction results of a row in the camera image are extracted and fitted into a line. The error is shown in Figure 8c. The raised and recessed parts near the square boundary are obviously eliminated.

Secondly, the metal plane in Figure 4a was also reconstructed. The area with specular reflectance was reconstructed and analyzed in Figure 9 and Table 1. Due to specular reflectance, the reconstructed surface includes wave-like unevenness. After taking the reflectance properties into consideration, a smooth surface can be acquired by our proposed method. In detail, after the specular components were subtracted from the original intensity, only the diffuse components were used to calculate the subpixel location of the stripe. We observe that while specular reflectance degrades the performances of the method in [2], our method works better since specular reflection was eliminated first in the process of stripe detection. Our proposed method can improve the robustness of stripe detection and reduce the reconstruction error for shiny surface. The metal plane was also reconstructed by projecting sinusoidal fringe patterns. As shown in Figure 9c,d for the over-exposed region, the obvious errors can be observed with the maximum error 0.4137 mm and the minimum error −0.4438 mm. For a metal plane, the overall error within 0.89 mm can be observed. The smaller error 0.5247 mm by a stripe-based structured light method shows that the stripe-based structured light is more robust than the phase-shifting-based method. In summary, our method acquired the smallest reconstruction error (0.2158 mm).

The maximum, minimum and deviation error for the above two objects were listed in Table 1.

### 5.2. Free-Form Object

In this part, non-Lambertian surface with specular region and texture, including free-form metal object and a vase, were reconstructed as well. To compare the reconstruction results quantitatively, the objects were also scanned by a laser tracker measurement system and the reconstructed results were used as criteria and compared with ours. Due to the higher reconstruction resolution of our results, the reconstructed results by our proposed method were first down-sampled and registered. Then the maximum and minimum error and the standard deviation were used to evaluate the accuracy of several methods. The statistical results are listed in Table 2. Based on the quantitative comparisons, we can find that our proposed method acquired the smallest reconstruction error, which validates the effectiveness of our proposed method on eliminating the effect of reflectance properties on reconstruction in a stripe-based structured light system.

As shown in Figure 10a, a vase with a colored texture was reconstructed. The diffuse component was first calculated and the visual albedo map was shown in Figure 10b. The reconstruction results and area enlarged by the method in [42] are shown in Figure 10c,e. Compared with our results in Figure 10d,f obvious unevenness near the texture caused by the colored texture can be observed. In addition, due to the ceramic material, wave-like unevenness can be observed as well. Our proposed method acquired more smooth results.

Finally, a free-form metal object was reconstructed, and the shiny area can be observed in the all-white image in Figure 11a. Firstly, to show the capacity of the Blinn-Phong model to represent the reflectance properties, the restored image of the all-white image was shown in Figure 11b based on the estimated parameters. The results of stripe detection by ours were also given in Figure 11c. The numbers “128” and “255” represent the rising or falling edge of the detected stripe in the normal pattern. Due to the high-reflective area, the detection results by method in [2] lacks robustness and near some high-reflective area, the stripe detection failed, which results in a poor reconstruction results as shown in Figure 11e. By extracting the diffuse components from the original intensity based on the estimated reflectance parameters, an accurate result of detection and reconstruction can be acquired as shown in Figure 11c,f. In addition, the reconstructed model by the three-step phase-shifting algorithm was shown in Figure 11d. The saturated region was over-exposed and the coding information in sinusoidal fringe pattern was inaccurate or missed, which leads to apparent hole and reconstruction errors. More robust reconstruction results can be acquired by our proposed method which validates the effectiveness and improvement on the existing 3D reconstruction technique.

## 6. Conclusions and Future Work

In this paper, we proposed a hybrid system consisting of stripe-based structured light and photometric stereo. Based on the conclusion that the reconstruction quality is strongly dependent on the interplay of surface geometry, illumination and surface reflectance, we proposed an improved stripe detection method. Firstly, based on the error analysis on stripe-based structured light method, the Blinn-Phong model was used in a photometric stereo system to divide the reflected intensity into two components, including the diffuse component and the specular reflection component. The nonlinear relationship between the incident intensity and the reflected intensity for each camera pixel was established. Then in the process of stripe detection, the specular component and diffuse albedo was eliminated to reduce the effect of the surface texture and shiny surface on the stripe detection. Higher-accuracy and more robust reconstruction results for shiny objects and textured surfaces by our proposed method can be acquired. In addition, the three-step phase-shifting method was implemented and the reconstruction results and discussion were given and compared with ours. The results show that stripe edges generally can be better preserved than individual image intensity in the presence of complex reflection, as binary stripe coding strategy combined with stripe subpixel detection is more robust. In experimental parts, a metal-flat and checkerboard were reconstructed to validate our proposed method. Free-form objects, including metal objects, a vase with a colored texture and a porcelain plate with Chinese characters, were reconstructed and compared with existing stripe-based and phase-shifting structured light methods. The results show that improved reconstruction results by our proposed method can be acquired and further used to combine with normal information for detail-preserving. Fusion methods combining the depth values and normal information have been concluded in Section 2. In the future, we can adjust the location of the projector lying in the same circle with area light sources. In this way, more accurate non-Lambertian properties can be acquired by the photometric stereo method so that more accurate stripe detection can be acquired. In addition, a more general reflectance model applicable for general objects will be used to enlarge the application range of our hybrid method. As the 3D measurement of complex scenes with varying reflections is an ongoing and never-ending problem, it is hoped that our method can provide new ideas and perspectives for solving such problems.

## Figures and Tables

**Figure 1 sensors-20-06564-f001:**
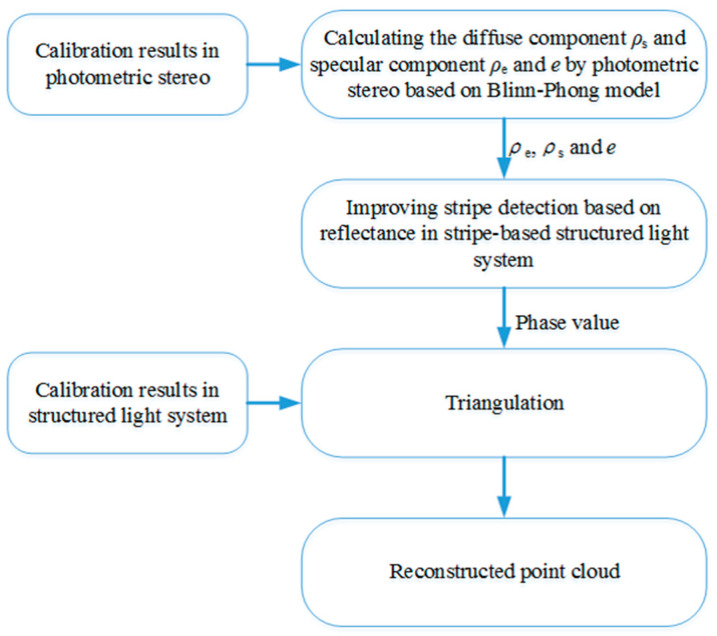
Flowchart of the proposed 3D measurement system.

**Figure 2 sensors-20-06564-f002:**
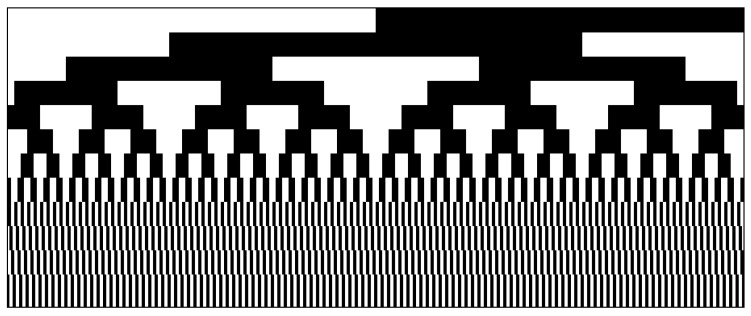
Coding strategy of Gray code and stripe shifting pattern. Top: Series of Gray code (*n* = 8) divide region into 256 subregions. Bottom: Stripe pattern with width of 4 pixels is shifted three times to encode positions within each subregion.

**Figure 3 sensors-20-06564-f003:**
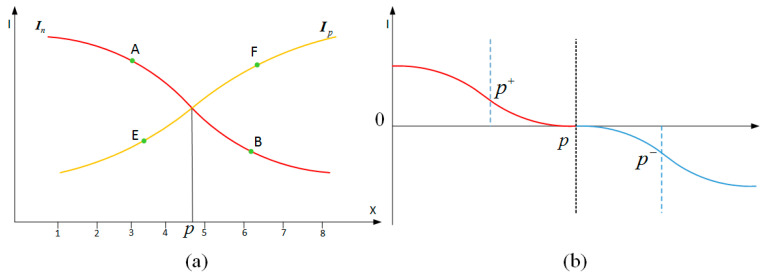
Stripe detection methods. (**a**) Using normal and inverse stripe pattern; (**b**) The improved detection method for shiny surface.

**Figure 4 sensors-20-06564-f004:**
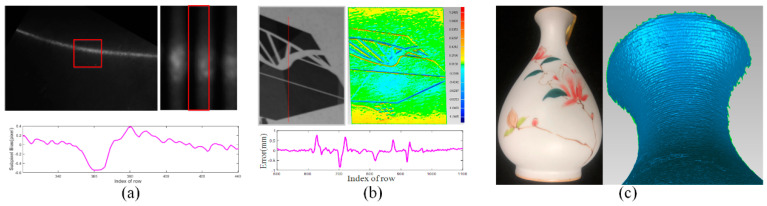
Example of biased detection. (**a**) The detection bias is introduced due to specular reflection; (**b**) The detection bias is introduced due to surface texture; (**c**) The apparent unevenness due to glossy reflection and surface texture.

**Figure 5 sensors-20-06564-f005:**
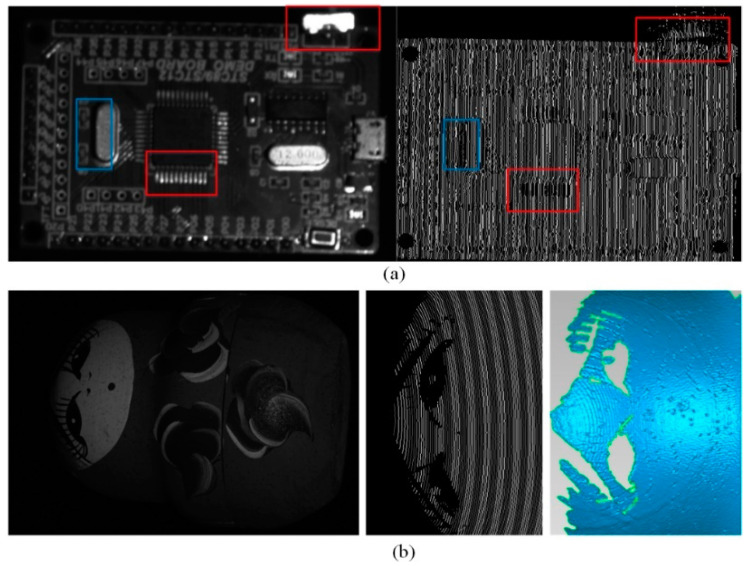
Example of missed detection. (**a**) Specular reflection and geometric occlusion result in missed detection; (**b**) Region with black texture leads to missed detection.

**Figure 6 sensors-20-06564-f006:**
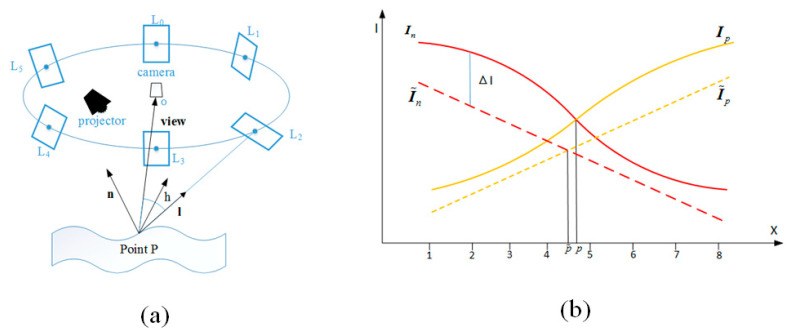
(**a**) Illustration of reflectance model. Apart from incoming light, the intensity is related to the direction of view and the halfway vector ***h***. *L*_0_~*L*_5_ is the number of the area light source; (**b**) Illustration of our proposed method. *In* and *Ip* are the original intensity vector of the detected stripe; ${\tilde{I}}_{p}$
and ${\tilde{I}}_{n}$ are the improved ones with the modulation resulting from geometry only.

**Figure 7 sensors-20-06564-f007:**
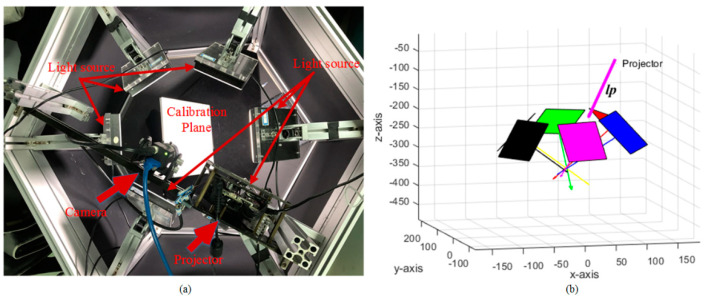
Hardware and calibration results. (**a**) Our hardware consists of a camera, a projector and six area light sources; (**b**) The illustration of calibration results; the length and direction of arrow represents the intensity and direction of light source.

**Figure 8 sensors-20-06564-f008:**
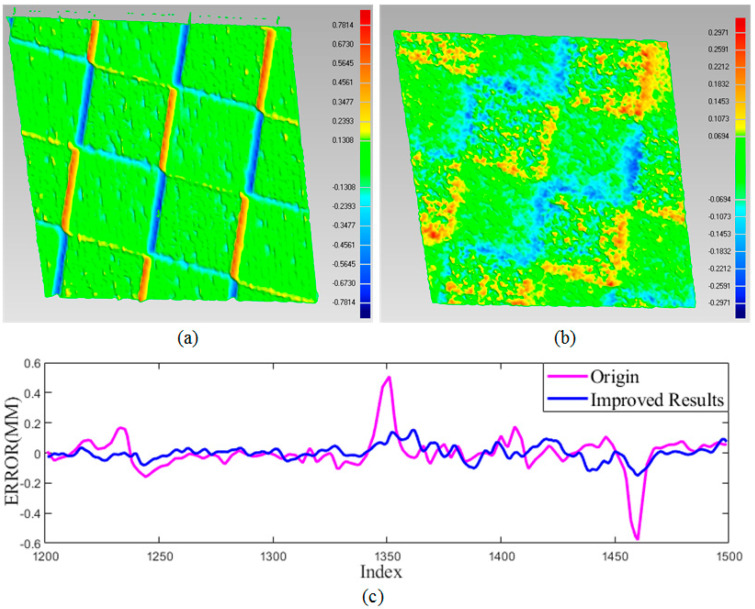
Comparisons and results of a checkerboard. (**a**) The result by [42]; (**b**) The result by ours; (**c**) The errors from the same row in the camera image.

**Figure 9 sensors-20-06564-f009:**
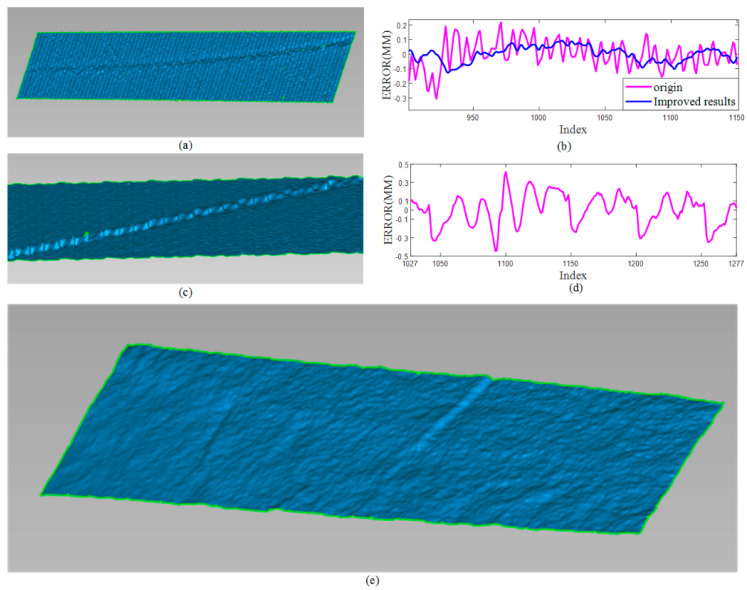
Results and comparison of metal plane. (**a**) The result by [2]; (**b**) The errors from the same row in camera image; (**c**) The result by phase-shifting method in [28]; (**d**) The errors from the same row by phase-shifting method in [28]; (**e**) The results by ours.

**Figure 10 sensors-20-06564-f010:**
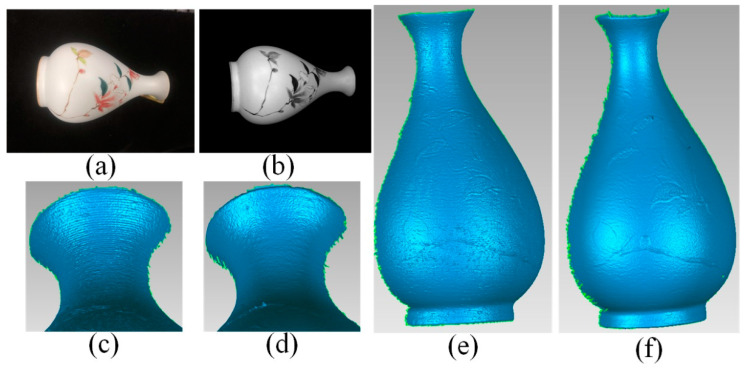
Results and comparisons of a vase. (**a**) A vase with colored texture and glossy reflection; (**b**) The visual albedo map; (**c**) Local areas by [42] are enlarged; (**d**) Local areas by ours are enlarged; (**e**) The reconstruction results by [42]; (**f**) The reconstruction results by ours.

**Figure 11 sensors-20-06564-f011:**
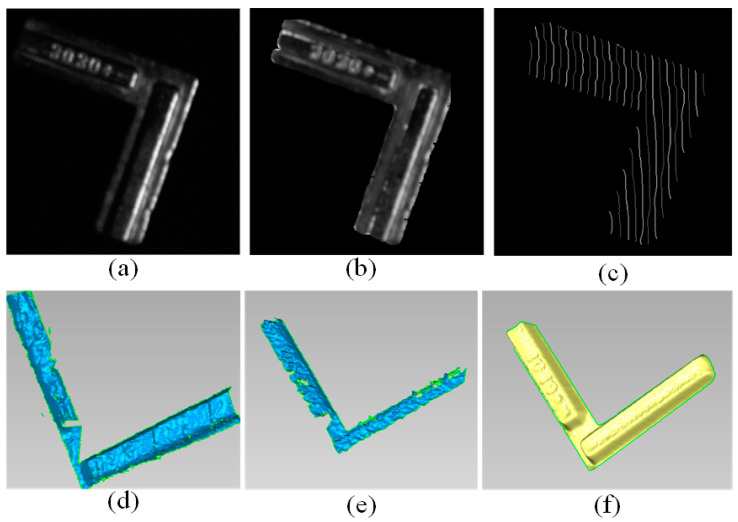
Results and comparisons of a metal object. (**a**) A metal object with specular reflection; (**b**) The restored all-white image based on the estimated reflectance parameters; (**c**) The result of stripe detection by ours; (**d**) The reconstruction result by phase-shifting method in [28]; (**e**) The reconstruction result by [2]; (**f**) The reconstruction result by ours.

**Table 1 sensors-20-06564-t001:** Comparisons of reconstruction results (mm).

Category	Method	Maximum Error	Minimum Error	Std
Checkerboard	[42]	0.5077	−0.5781	0.1148
	Ours	0.1511	−0.1506	0.0502
Metal Plane	[2]	0.2186	−0.3061	0.0911
	[28]	0.4137	−0.4438	0.1684
	Ours	0.0929	−0.1269	0.0520

**Table 2 sensors-20-06564-t002:** Comparisons of reconstruction results of the free-form surface (mm).

Category	Method	Maximum Error	Minimum Error	Std
**Vase**	[42]	3.5426	−2.4502	0.5369
	Ours	0.5642	−0.3254	0.1124
**Free-form metal**	[28]	2.3659	−2.2132	0.6985
	[2]	1.5368	−0.9354	0.2365
	Ours	0.6586	−0.5252	0.1352

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
