# Peer review of "Eliminating the Effect of Reflectance Properties on Reconstruction in Stripe Structured Light System"

_sensors, 2020, doi:10.3390/s20226564_

Round 1

Reviewer 1 Report

The aim of this paper was to presented the hybrid system consisting of structured light technique and photometric stereo combines the depth value with normal information to refine the reconstruction results. The presented investigation is very interesting, nevertheless,  authors should consider the following issues:

  1. The main drawback of this investigation is lack of comparing with the ground truth data obtained from i.e. coordinate machine or laser tracker.
  2. Visual inspection without extended statistical analysis did not allow for full quality assessment.
  3. 1 – it is hard to understand (based on only the figure) how the calibration was performed. Also to localisation of the Figure in opinion should be moved near to the main text (i.e. line 83).
  4. Line 99 Please explain what kind of simple logical operations and tools from combinatorial mathematics were used.
  5. Similar to the Fig 1, Fig. 3 should be placed in another place in the text.
  6. Section 3.1 Please add more information to the quality assessment section. Without the GSD size it is hard to assets if the obtained values are satisfied. Please also add more detail to the glossy vase.

Kind regards,

Reviewer

Author Response

Thanks for your time and patience. Based on our summary of the editor and all reviewers' comments, three parts of the content have been modified in the revised manuscript as follows:

1. Quantitative comparisons and discussion on free-form objects including the vast, porcelain plate and the free-form metal object.                    

1.1 The GSD sizes of the vast, porcelain plate and the free-form metal object were acquired by laser tracker.        

1.2 The maximum, minimum error and the standard deviation were used to evaluate the accuracy of several   methods. The statistical results were listed in Table 2 in our revised manuscript. The related discussion have been added in the revised manuscript.

2. We have relocated all figures as the reviewer suggested.

3. Three problems from the editor have been solved as follows.                                                       

3.1 We have revised the authors order in the manuscript consistent with that in the system. 

3.2 All the figures with copyright issues have been replaced with our own figures.               

3.3 The Chinese characters in Figure 10 have been removed as well.

The remaining questions and concern of the reviewers are addressed carefully in detail as below.

Wishing you good health during this difficult time.

Sincerely

Reviewer 2 Report

This paper proposed a hybrid structured light system for surface texture and non-Lambertian depth sensing. The novel stripe detection method can deal this problem well. And the results have proved the performance of the method. However, there are some comments need to be considered: (1) As shown in Fig.2, the author said that minimum stripe width is significant. However, the minimum stripe tends to blur when this pattern is projected. How to anti this blur? Meanwhile, the narrow of the stripe affects the projected energy of the light. (2) What's the parameter of I0'(i,j) in Eq.(1)? (3)The calibration of the reflectance model is unique or changes with the material of the scene? (4)Some latest gray code structured light method should be referenced. Eg.1 Single-Shot Colored Speckle Pattern for High Accuracy Depth Sensing; Eg.2 Single-Shot Dense Depth Sensing with Color Sequence Coded Fringe Pattern.

Author Response

(The authors gave the same response as above.)

Round 2

Reviewer 1 Report

Dear Authors,

Thank you very much for your response. Adding the statistical analysis and ground truth data obtained by a laser tracker allows to perform correct quality assessment

One thing that authors should consider is change the word error (i.e. max/min error) to  deviation (maxim/minimum).

After correction of this small errors, article will acceptable for publishing.

Kind regards,

Reviewer

Author Response

Thanks for your suggestion. We have corrected the word error in our revised manuscript.